# Protective effect of sucrose esters from cape gooseberry (*Physalis peruviana* L.) in TNBS-induced colitis

Yanet C. Ocampo[1], Jenny P. Castro[1,2], Indira B. Pájaro[1,2], Daneiva Caro[1], Elena Talero[3], Virginia Motilva[3], Luis A. Franco[1]*

1 Biological Evaluation of Promising Substances Group, Faculty of Pharmaceutical Science, Universidad de Cartagena, Cartagena, Colombia, 2 Faculty of Chemistry and Pharmacy, Universidad del Atlántico, Barranquilla, Colombia, 3 Department of Pharmacology, Facultad de Farmacia, Universidad de Sevilla, Sevilla, Spain

* lfrancoo@unicartagena.edu.co

**Data Availability Statement:** All data files are available from the Universidad de Cartagena repository (https://repositorio.unicartagena.edu.co/handle/11227/5014.).

## Abstract

Phytotherapy is an attractive strategy to treat inflammatory bowel disease (IBD) that could be especially useful in developing countries. We previously demonstrated the intestinal anti-inflammatory effect of the total ethereal extract from the *Physalis peruviana* (Cape gooseberry) calyces in TNBS-induced colitis. This work investigates the therapeutic potential of Peruviose A and B, two sucrose esters that constitute the major metabolites of its calyces. The effect of the Peruvioses A and B mixture on TNBS-induced colitis was studied after 3 (preventive) and 15-days (therapy set-up) of colitis induction in rats. Colonic inflammation was assessed by measuring macroscopic/histologic damage, MPO activity, and biochemical changes. Additionally, LPS-stimulated RAW 264.7 macrophages were treated with test compounds to determine the effect on cytokine imbalance in these cells. Peruvioses mixture ameliorated TNBS-induced colitis in acute (preventive) or established (therapeutic) settings. Although 3-day treatment with compounds did not produce a potent effect, it was sufficient to significantly reduce the extent/severity of tissue damage and the microscopic disturbances. Beneficial effects in the therapy set-up were substantially higher and involved the inhibition of pro-inflammatory enzymes (iNOS, COX-2), cytokines (TNF-α, IL-1β, and IL-6), as well as epithelial regeneration with restoration of goblet cells numbers and expression of MUC-2 and TFF-3. Consistently, LPS-induced RAW 264.7 cells produced less NO, PGE2, TNF-α, IL-6, and MCP-1. These effects might be related to the inhibition of the NF-κB signaling pathway. Our results suggest that sucrose esters from *P. peruviana* calyces, non-edible waste from fruit production, might be useful as an alternative IBD treatment.

## 1. Introduction

Inflammatory bowel disease (IBD), including Crohn's disease and ulcerative colitis, represents a group of chronic, relapsing, and remitting inflammatory conditions of the gastrointestinal tract that affect millions of people worldwide [1]. The incidence of IBD remains relatively

**Funding:** This work was supported by Colciencias, actually Minciencias [Grant 110751929179–2010 and 597-2012], and the Universidad de Cartagena (Grant 055-2013). The funders had no role in study design, data collection and analysis, decision to publish, or preparation of the manuscript.

**Competing interests:** The authors have declared that no competing interests exist.

constant in regions like Northern Europe and North America. Conversely, the time-trend studies suggest that IBD rises in areas where incidence was previously low, such as Southern Europe, Asia, Africa, and South America [2, 3]. There is no cure for IBD; however, a variety of treatments are available, including aminosalicylates, corticosteroids, thiopurines, methotrexate, cyclosporine, and biologics such as anti-TNFα therapies [4, 5]. Despite this, current IBD therapies have important shortcomings regarding safety, efficacy, and applicability. Their potentially serious side effects remain a major clinical problem and limit their chronic use. On the other hand, the lack of effectiveness is translated into the fact that a large percentage of IBD patients will require intestinal surgery to treat complications at some point following diagnosis [6]. Therefore, novel, safe, and effective therapeutic strategies are highly desirable.

In this context, the use of herbal medicine to treat IBD is increasing and gaining popularity globally, mainly because of its low toxicity, biocompatibility, and affordability [6, 7]. This is particularly important in developing countries, where the access of patients to specialized centers and standard therapies is limited [8]. Various natural products have been shown to safely suppress pro-inflammatory pathways and control IBD in both pre-clinical and clinical studies [9–11]. Consequently, the evaluation of extracts from plants traditionally used to treat inflammatory conditions is an important approach for developing future therapies for IBD. However, progress in herbal therapeutic research and its clinical application is limited by the lack of identification of bioactive compounds, insufficient preclinical studies using animal models, and unclear mechanism of action [12–14].

*Physalis peruviana* L. (Solanaceae), commonly known as cape gooseberry, is native to South American countries, including Colombia, Ecuador, Brazil, and Peru [15]. In Colombia, cape gooseberry is one of the most important exotic fruit crops, representing a source of profit for some regions of the country [15, 16]. For more than 20 years, Colombia has been at the forefront of the world production of cape gooseberry. The area planted with this crop increased significantly in the last years, reaching 1,311 hectares in 2018 [16, 17]. In 2021, Colombia came to the highest figure for cape gooseberry exports, with an increase in the number of tons exported, making it the world's largest producer and exporter of this fruit [18]. *Physalis peruviana* is characterized because its fruit grows enclosed in an inflated calyx, formed by five persistent sepals that develop a 5 cm-size structure that resembles a Chinese lantern. Calyces protect the fruit against insects, pathogens, or adverse climates and play a major role in fruit development, both in size and carbohydrate content [19]. Calyces, as well as leaves and small stems, are widely employed as traditional herbal remedy medicine for their anti-inflammatory properties [15].

Despite the folk employment of calyces to treat inflammation, the fruit is still perceived as the only valuable part of the plant, useful for nutritional or medicinal purposes. Indeed, cape gooseberry is highly appreciated on the international market, particularly in France and Germany, where the fruit is considered a delicacy for its unique flavor, texture, and color [20]. In this sense, calyces from *Physalis peruviana* are merely employed as decoration for desserts, meals, and salads or to increase the shelf life of the fruit after harvesting for exportation to European countries [21, 22]. Otherwise, calyces are removed and discarded, representing most of the waste generated during cape gooseberry production.

In a previous study, we demonstrated that the total ethereal extract from the calyces of *Physalis peruviana* significantly ameliorates intestinal inflammation in a murine model of IBD by reducing the production of TNF-α and IL-1β levels [23]. These results established the ability of *Physalis peruviana* calyces to suppress the altered colon immune response. On the other hand, we found that the major anti-inflammatory metabolites from this ethereal extract were a mixture of sucrose esters, Peruvioses A and B. These compounds presented a higher anti-inflammatory activity than their parental extract at lower and well-tolerated doses [15, 24]. Thus, in this study, we explored the potential of sucrose esters from *Physalis peruviana* to modulate the

altered inflammatory response in IBD. For this, we employed the 2,4,6-trinitrobenzene sulfonic acid (TNBS)-induced colitis model in rats in two different set-ups: preventive and therapeutic. In addition, we investigated the effect of the Peruvioses mixture on cytokine production by LPS-activated macrophages and their antioxidant capacity *in vitro*.

## 2. Material and methods

### 2.1 Plant material

*Physalis peruviana* L. was collected in La Mesa, Colombia (4˚37′ 49.22′′ N; 74˚27′ 45.60′′ W; elevation 1198 m.a.s.l.) in November 2003. A voucher specimen (COL-512200) has been deposited at Herbario Nacional Colombiano (Instituto de Ciencias Naturales, Universidad Nacional de Colombia), Bogotá, Colombia.

### 2.2 Sucrose esters isolation

Isolation of Peruviose A and B from *P. peruviana* was carried out as previously described [15]. Briefly, dried calyces (2 Kg) were powdered and extracted with petroleum ether until exhaustion. The total extract (265 g) was partitioned with ether and methanol-water (9:1), and part of the hydroalcoholic fraction (20 g) was further purified by column chromatography on silica gel. The major fraction (8.23 g) was purified by preparative TLC to yield 1100 mg of a mixture of Peruviose A and B, occurring in a 6:4 ratio, respectively. Purity of the sucrose esters mixture was estimated to be 99.05 % by HPLC (LaChrom Elite®, Merck-Hitachi, Darmstadt, Germany), and their structures were confirmed by comparison of their IR and [1]HNMR data with those previously reported [15]. For details, see Supplementary Information.

### 2.3 Animals

Six to seven-week-old female Wistar rats (149–195 g) provided by Instituto Nacional de Salud (Bogota, Colombia) were housed in filtered-capped polycarbonate cages containing wood shavings. They were maintained under a 12 h light/darkness cycle in a temperature (22±3 ˚C) and humidity (65–75%) controlled environment. Animals were allowed to acclimatize for two weeks before use and fed with standard rodent food and water *ad libitum*. Rats were deprived of food 12 h prior to the induction of colitis but were allowed free access to water throughout. They were randomly assigned to groups of eight to twelve animals in a blinded fashion. Experiments followed a protocol approved by the local Ethics Committee of the University of Cartagena (Minutes of October 23, 2010), and all experiments were in accordance with the recommendations of the European Union regarding animal experimentation (Directive of the European Council 2010/63/EU).

### 2.4 Sucrose esters treatment

Given the low solubility of the mixture of Peruviose A and B from *P. peruviana* in water, sucrose esters were co-precipitated with polyvinylpyrrolidone (PVP K-30, USP grade) in a 1:4 (w/w) ratio by solvent evaporation technique. In brief, 6 g of Peruvioses mixture was dissolved in 30 mL of dichloromethane and mixed with 24 g of PVP. The resultant solution was dried with a rotatory evaporator with reduced pressure and controlled temperature (40±5˚C). Final solvent removal was done after manually gently scraping the co-precipitate and drying it in the oven at 40±5˚C overnight. The dried samples were stored in a desiccator at room temperature until use. Sucrose esters were solubilized in phosphate saline buffer (PBS, pH 7.3) and administered intraperitoneally (ip) daily at different doses. The duration and doses of sucrose esters treatment depended on each experiment's setting, as detailed below.

## 2.5 Induction of TNBS colitis

Colitis was induced according to the procedure described by Morris *et al.* (1989) [25], with some modifications. Rats were fasted for 12 h and anesthetized with a mixture of 100 mg ketamine (Lab Biosano, Santiago, Chile) and 5 mg of diazepam (Viteco S.A., Bogotá, Colombia) per Kg of body weight. Subsequently, 0.25 mL of TNBS (40 mg/mL; Sigma-Aldrich, St Louis, MO, USA) dissolved in ethanol (50% v/v) was instilled rectally using a cannula (external diameter 2 mm) introduced to an 8 cm depth. A healthy control group was included, which received saline in a comparable volume. Following the instillation of the hapten, the animals were maintained in a head-down position until they recovered from anesthesia to prevent leakage of the intracolonic instilled.

Two different protocols were designed to study the anti-inflammatory effect of the Peruvioses A and B mixture. In the first approach, rats were preventively treated with sucrose esters (20, 10, and 5 mg/kg/day, ip) 48, 24, and 2 h before colitis induction as well as 24 h thereafter, and were sacrificed 72 h after colitis induction. To evaluate the therapeutic effect, sucrose esters (10 and 5 mg/kg/day, ip) were administered after TNBS instillation and daily for two weeks before animal sacrifice. Control and TNBS groups were also treated with vehicle (PVP K-30 dissolved in PBS) by ip route. In both outlines, animal body weight, the occurrence of diarrhea, and food intake were monitored daily. Once the animals were sacrificed by cervical dislocation, the colon was removed, cleaned, and longitudinally opened, and its weight/length ratio was recorded. Each colon was scored for macroscopically visible damage by measuring the extent (cm$^2$) and severity of the lesions (score 0–10) in the distal colon as well as the presence of adhesions (score 0–2), according to the criteria of Bobin-Dubigeon *et al*, 2001 [26]. Representative colon, liver, and kidney samples were collected and fixed in buffered formalin for the histological analysis. Pieces of the colon were collected and either placed in RNA Later® (Qiagen, Valencia, CA, USA) or frozen in liquid N$_2$ immediately on the collection and stored at -80˚C until the measurement of biochemical parameters or RNA extraction was performed.

## 2.6 Histology analysis

Paraffin-embedded colon samples were sliced at 5 μm and stained with hematoxylin and eosin (H&E), or periodic acid Schiff (PAS), according to standard protocols, for histological evaluation of colonic damage and mucus content. Samples were blindly analyzed by an experienced pathologist employing light microscopy (Olympus BX41, Tokyo, Japan). The histological damage was evaluated using the scoring system described by Arribas *et al*, 2010 [27]. The extent of mucus production was estimated by determining the number of PAS-positive goblet cells in at least three fields of 23.54 mm$^2$, randomly selected from each slide at 20X magnification. Liver and kidney samples were stained with H&E and evaluated according to the parameters described in Supplementary S1 Table. Images were captured at 10X and 40X magnifications on a Zeiss Axio Lab.A1 microscope (Carl Zeiss, Oberkochen, Germany), coupled to a computer-driven Zeiss AxioCAM digital camera (ICc5).

## 2.7 MPO activity assay

MPO activity was measured as an index of neutrophil infiltration according to the method of Grisham *et al*, 1990 [28]. Colon tissue was homogenized in phosphate-buffered saline (PBS, pH 6.0) containing 0.5% hexadecyl-trimethylammonium bromide (HTAB) and subjected to cycles of freezing/thawing and brief periods of sonication (15 s). The homogenates were centrifuged for 10 min at 4000 rpm and 4˚C, and a sample of supernatants (20 μL) was mixed with 50 μL of *O*-dianisidine dihydrochloride (0.067%) and 50 μL of hydrogen peroxide (0.003%)

and incubated for 5 min. Enzyme activity was determined by measuring changes in the optical density at 450 nm ($OD_{450}$), using a microplate reader (Thermo Scientific, Waltham, MA, USA), and expressed as enzyme activity units, where an activity unit is defined as the amount of enzyme capable of metabolizing 1 μmol of hydrogen peroxide in one minute at 25˚C. The results are expressed as MPO activity per mg of tissue.

## 2.8 Measurement of cytokine levels

Colon biopsies were homogenized in Greenburger buffer pH 7.4, containing protease inhibitors (cOmplete Mini EDTA free; Roche, Basel, Switzerland), sonicated for 10 sec, and centrifuged at 10.000 rpm and 4˚C for 10 minutes. Homogenates were stored at -20˚C until use. Levels of IL-1β, IL-4, IL-6, IL-10, INF-γ, and TNF-α, were quantified by ELISA (Invitrogen, Carlsbad, CA, USA) according to the manufacturer protocol. Cytokine levels were normalized using the total quantity of proteins determined by the Bradford method (Bio-Rad, Hercules, CA, USA).

## 2.9 Quantitative real-time PCR (RT-PCR)

Total RNA was isolated from colon tissue using the RNeasyⓇ kit (Qiagen, Valencia, CA, USA) as described by the manufacturer. RNA was quantified and purity assessed by determining the 260/280 nm absorbance ratio using a NanoDrop 2000c spectrophotometer (Thermo Scientific, Waltham, MA, USA). For each sample, 2.0 μg of RNA was employed as template for cDNA synthesis using the Quantitect reverse transcription kit (Qiagen, Valencia, CA, USA). Real-time PCR analysis was performed using the LightCyclerⓇ 96 System (Roche, Mannheim, Germany) with Power SYBRⓇ Green PCR master mix (Applied Biosystems, Forsters, CA, USA) and specific primers (Eurofins Genomics, Huntsville, AL, USA) according to the manufacturer's instructions. Primer sequences were obtained from the literature and tested for sequence specificity using the Basic Local Alignment Search Tool at NCBI (http://blast.ncbi.nlm.nih.gov). Gene names, accession numbers, and primer sequences are listed in S2 Table. Gene expression was normalized to glyceraldehyde-3-phosphate dehydrogenase (GAPDH) as endogenous housekeeping gene. Duplicate cycle threshold (CT) values were analyzed by the comparative CT (ΔΔCT) method.

## 2.10 Western blot

Colon tissue was homogenized, and cytoplasmic/nuclear proteins were isolated with the ReadyPrep protein extraction kit (Biorad, Hercules, CA, USA) according to the manufacturer´s instructions. Protein concentration was determined by the Bradford method (Bio-Rad, Hercules, CA, USA). Aliquots of 30 μg proteins were fractionated by SDS-PAGE (7.5% tris-glycine-polyacrylamide gels) and transferred to ImmobilonⓇ-P membrane (Millipore, Bedford, MA, USA). The membrane was blocked with 5% milk in 0.1% Tween-20-PBS for 1 h and incubated overnight with primary antibodies. Antibodies for β-actin (sc-47778), iNOS (sc-7271), and NF-κB p50 (sc-166588) were purchased from Santa Cruz Biotechnology (Santa Cruz, CA, USA) and used at a dilution of 1:200 to 1:500. The membranes were washed with 0.1% Tween 20-PBS and then incubated for 1 hour with the appropriate horseradish peroxidase-conjugated secondary antibody (Santa Cruz Biotechnology, Santa Cruz, CA, USA), washed again and incubated with chemiluminescent substrate (Clarity™ Western ECL Substrate; Bio-Rad, CA, USA) for 5 min. The membranes were captured using the G: BOX imager (Syngene, Cambridge, UK), and protein bands quantified with the GeneTools analysis software (Syngene, Cambridge, UK).

## 2.11 Cell culture

RAW 264.7 macrophages were obtained from the American Type Culture Collection (TIB-71; Rockville, MD, USA) and routinely cultured in Dulbecco's Modified Eagle Medium-high glucose (DMEM) supplemented with 2 mM L-glutamine, antibiotics (100 UI/mL of penicillin-100 μg/mL streptomycin) and 10% heat-inactivated fetal bovine serum at 37˚C in a humidified atmosphere containing 5% $CO_2$ and 95% air.

## 2.12 MTT reduction assay

Cytotoxic effect of test compounds was evaluated using the 3-(4,5-dimethylthiazol-2-yl)-2,5-diphenyltetrazolium bromide test (MTT assay) [29]. RAW 264.7 cells were cultured into 96-well plates ($2x10^5$ cells/mL) and allowed to grow to confluence. Subsequently, the culture medium was discarded and replaced with sucrose esters (0–100 μg/mL) or ethanol vehicle for 30 minutes, followed by stimulation with 10 μg/mL lipopolysaccharide (LPS). After 24 h, the supernatant was replaced by fresh medium containing MTT (0.2 mg/mL). Four hours later, the medium was carefully aspirated, and formazan crystals were dissolved in DMSO (100 μL), and the $OD_{550}$ was measured using a microplate reader. The viability of treated cells was expressed as the percentage of control cells, which was assumed to be 100%.

## 2.13 Macrophages stimulation assay

RAW 264.7 macrophages were treated like that described for cell viability. In brief, cells were seeded in 24-well plates ($2x10^5$ cells/mL) and treated for 30 min with various concentrations of sucrose esters (0–20 μg/mL), N-(3-(aminomethyl)benzyl)acetamidine (1400W, 2.50 μg/mL), rofecoxib (3.14 μg/mL) or dexamethasone (3.92 μg/mL), and stimulated with LPS (10 μg/mL). Control cells were cultured under the same conditions but were not exposed to the effect of LPS. Twenty-four hours later, culture supernatants were collected and stored at -20˚C until use.

NO production was estimated from the accumulation of $NO_2^-$ in the medium using the Griess reagent, as described previously [30]. Briefly, equal volumes of supernatants and Griess reagent (100 μL) were mixed and incubated at room temperature for 5 minutes. The $OD_{550}$ of the samples was measured using a microplate reader. The amount of nitrite in the samples was calculated from a standard curve (0–200 uM) of sodium nitrite ($NaNO_2$). Levels of IL-1β, IL-6, MCP-1, PGE2, and TNF-α, in culture supernatants were determined using ELISA (R&D Systems, Minneapolis, MN, USA or eBioscience, San Diego, CA, USA) according to the manufacturer's instructions. Final results were expressed as pg of mediator/mL of supernatant.

## 2.14 Statistical analysis

All values are expressed as mean ± standard error of the mean (SEM) for each group. One-way analysis of variance (ANOVA), followed by the Tukey post hoc test, was used to determine differences between treatment groups. Kaplan–Meier analysis with log-rank statistics was performed in the survival curve during TNBS-induced colitis. Values of $P<0.05$ were considered significant.

## 2.15 Ethical statement

This study was carried out in strict accordance with the recommendations of the European Union regarding animal experimentation (Directive of the European Council 2010/63/EU). The protocol was approved by the Committee of Ethics in Research of the University of Cartagena (Minutes of October 23, 2010). All efforts were made to minimize animal suffering.

# 3. Results

## 3.1 Intestinal anti-inflammatory effect of peruviose A and B in TNBS-induced colitis

In the preventive approach, Wistar rats were treated with the Peruviose A and B mixture from *Physalis peruviana* two days before rectal instillation of TNBS. Three days after colitis induction, the severity of intestinal inflammation was assessed by macroscopic, histological, and biochemical parameters (Fig 1). Rats treated with TNBS showed hypoactivity, piloerection, and diarrhea associated with reduced food intake (data not shown). Consequently, animals were severely anorexic with a substantial decrease in body weight in comparison to the control group ($P<0.0001$) (Fig 1B). Correspondingly, a macroscopic inspection of the colon showed evidence of severe inflammation characterized by necrosis of the mucosa, extending 5–10 cm$^2$ along the tissue, edema, hyperemia, deep ulcerations, and focal adhesion to adjacent organs. Although the pretreatment with sucrose esters did not modify food intake or body weight loss, compounds produced a significant anti-inflammatory effect when administered at doses of 10 and 20 mg/Kg/day, ip. As shown in Fig 1C, sucrose esters treated rats presented less severe and extended colon inflammation, showing a reduction of the macroscopic damaged area of at least 28.50% ($P<0.05$ vs. TNBS group). Conversely, the weight/length ratio of the rat colon

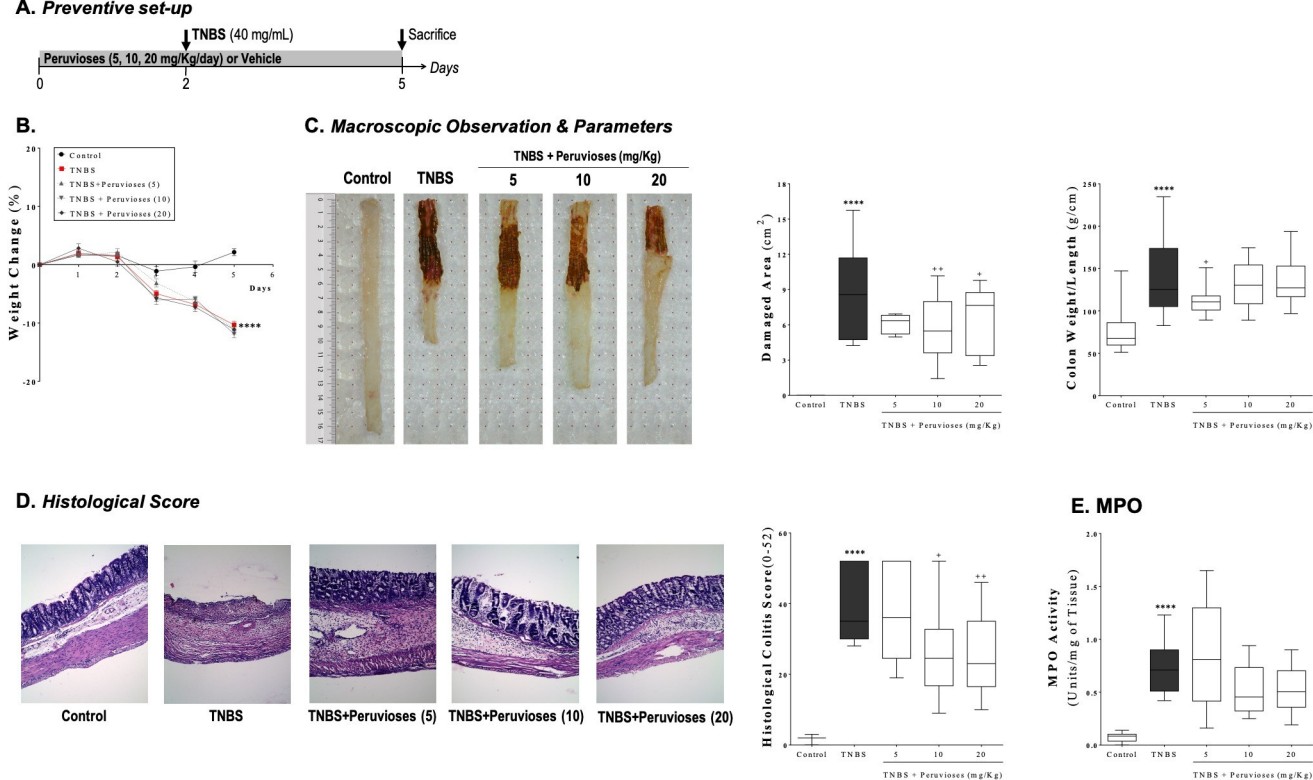

**Fig 1. Pre-treatment with Peruvioses A and B from *P. peruviana* ameliorates acute TNBS-colitis. (A)** Wistar rats were treated for 2 days with test compounds (5, 10, and 20 mg/Kg/day, ip) or vehicle. Colitis was induced by the instillation of TNBS. Three days after that, animals were sacrificed, and colitis severity was assessed. **(B)** Animals weight change was monitored daily. **(C)** Appearance, damaged area (cm$^2$), and colon weight/length ratio were evaluated as detailed in Materials and Methods. Representative pictures of rat colons are shown. **(D)** Histologic changes were examined by hematoxylin and eosin (H&E) staining and scored by a blinded pathologist. Representative pictures are shown (10X). **(E)** Myeloperoxidase (MPO) activity was measured in colon biopsies. Results represent at least two independent experiments and are expressed as the mean ± SEM. (n = 11–20 per group). (\*\*\*\*) $P<0.0001$ vs. control; (+) $P<0.05$ and (++) $P<0.01$ vs. TNBS group.

was not significantly reduced at these doses. Microscopically, colon tissue from the TNBS group was characterized by extensive ulceration and necrosis of mucosa, typically affecting over 75% of the surface, accompanied by massive depletion of goblet cells, transmural inflammation with edema and prominent infiltration of inflammatory cells at all the intestinal layers. In this group, the grade of the lesion was considered severe, with a score of 39.73±3.09 ($P<0.0001$ vs. control group, Fig 1D). Consistently, colonic injury induced by TNBS was characterized by an enhancement in myeloperoxidase (MPO) activity compared to the control group ($P<0.0001$, Fig 1E), suggesting extensive neutrophil infiltration. Treatment with the Peruvioses mixture (10 and 20 mg/Kg) significantly diminished the histological score by over 35% ($P<0.05$ vs. TNBS group, Fig 1D) with dose-dependent reduction of mucosal ulceration, edema, number of infiltrating cells, and restoration of the colonic architecture as well as slight improvement of goblet cells numbers. Despite these observations, we found that MPO activity was not diminished significantly (Fig 1E).

To further characterize the beneficial effect of sucrose esters in TNBS-induced colitis, rats were treated with Peruviose A and B (5 and 10 mg/Kg/day) for 2 weeks after TNBS instillation, in a therapeutic setup of established colitis (Fig 2A) [31]. TNBS-induced colitis produced signs of abdominal pain, hypoactivity, piloerection, and a mortality rate exceeding 37% ($P<0.05$ vs. control group). In contrast, animals that received the higher dose of sucrose esters (10 mg/Kg) showed a slight improvement in mortality rates (20%, Fig 2B). Immediately after TNBS instillation, rats developed diarrhea and a reduction of food intake that was accompanied by a striking loss of body weight. As expected, the body weight of the rats with colitis remained significantly lower than that of the control animals for the duration of the study ($P<0.05$, Fig 2C). On the other hand, five to six days after TNBS instillation, sucrose esters-treated animals started to recover from the weight loss, returning to baseline body weight by day eight and increasing it significantly by the end of the experiment ($P<0.05$, vs. TNBS group). We also monitored the effect of sucrose esters on food intake and found a trend of recovery consistent with changes in body weight (Data not shown). Thus, treatment with the Peruviose A and B mixture resulted in an improved survival rate and reduced the manifestations of TNBS-induced colitis. Macroscopic damage induced by TNBS was characterized by adhesion to adjacent organs and mild ulceration, with a damaged area extending 7–24 cm$^2$ along the tissue. Furthermore, we found severe bowel wall thickening, with tissue having a hard and rigid texture suggesting fibrosis, as well as visible strictures, occasionally associated with obstruction (Fig 2D). The treatment with sucrose esters from *P. peruviana* reduced the extension and severity of macroscopic damage induced by TNBS ($P<0.01$), producing a recovery of tissue elasticity, texture, and thickness, as recorded by blinded observers. Consistently, rat colons weight/length ratio was significantly diminished at both high and low doses (Fig 2E, $P<0.05$ vs. TNBS group).

At the histological level, colon tissue from the TNBS group presented epithelial ulceration, affecting on average more than 50% of the surface, and complete necrosis of mucosa in 40% of analyzed slides, as well as transmural inflammation with profuse infiltration of inflammatory cells, and hyperplasia of all the intestinal layers. The inflammatory process was associated with crypt hyperplasia and substantial goblet cell depletion. In this group, the grade of the lesion was considered severe, with a score of 44.00±2.98 ($P<0.0001$ vs. control group). As expected, MPO activity increased significantly compared to control animals ($P<0.001$). Treatment with the mixture of Peruviose A and B (5 and 10 mg/Kg) produced a significant recovery of the damage induced by TNBS, giving a total score of 15.43±3.93 and 16.67±9.31 (Fig 2G–2I, $P<0.01$), respectively, graded as mild by the pathologist. Most of the samples showed a remarkable restoration of the epithelial cell layer and crypt architecture, with less than 16% of analyzed tissue showing complete necrosis of mucosa. The transmural involvement,

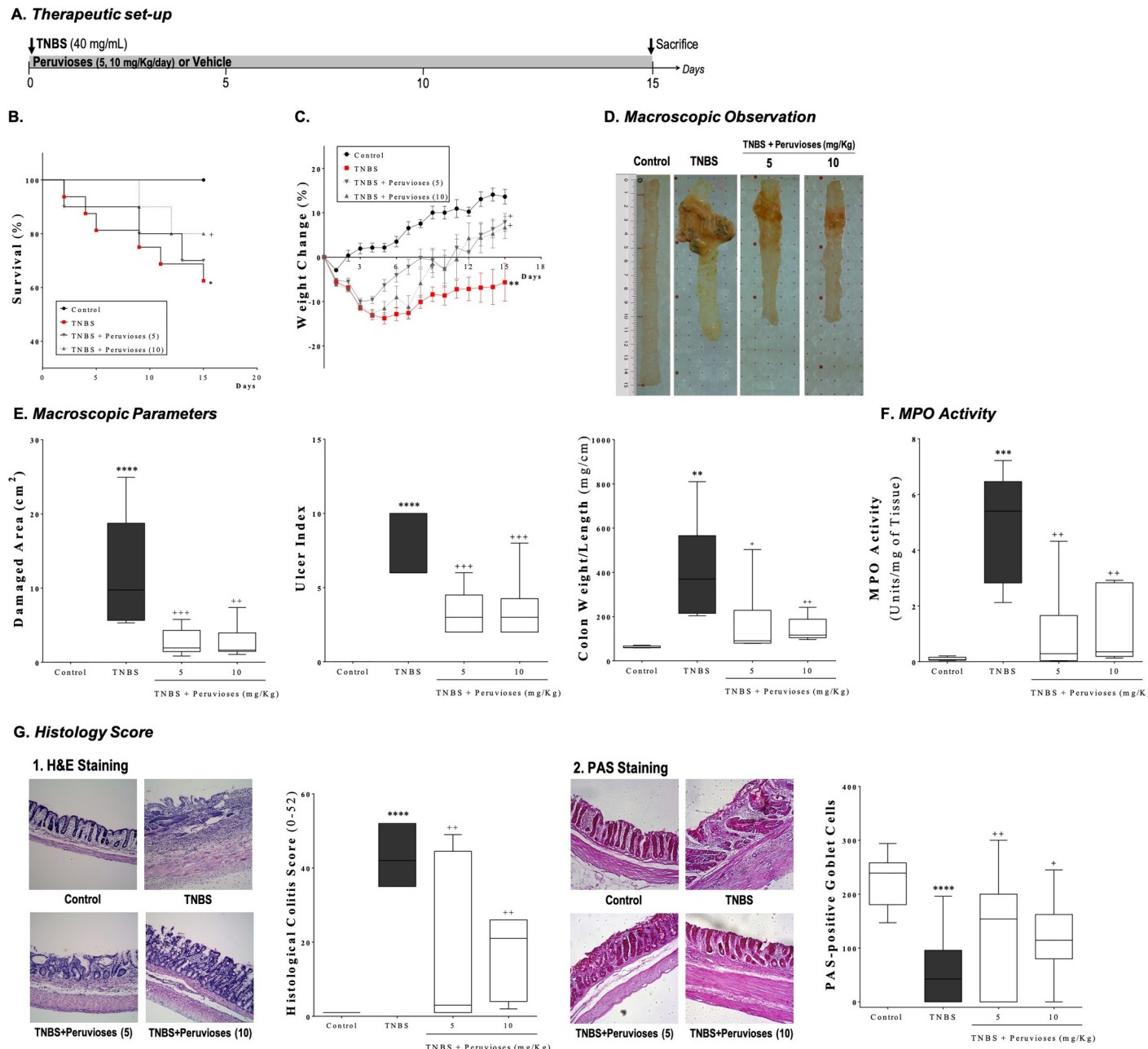

**Fig 2. Peruvioses A and B from *P. peruviana* diminished the inflammation induced by TNBS. (A)** Colitis was induced in Wistar rats by instillation of TNBS. Animals were treated with the Peruvioses mixture (5 and 10 mg/Kg/day, ip) or vehicle for 15 days. Afterward, rats were sacrificed, and colitis severity was assessed. **(B)** Survival and **(C)** body weight changes were monitored daily. **(D)** Appearance of colon tissue was evaluated, and representative pictures are shown. **(E)** Macroscopic damage (damaged area (cm$^2$), ulcer index, and colon weight/length) were scored as detailed in Materials and Methods. **(F)** Myeloperoxidase (MPO) activity was measured in colon biopsies. **(G)** Histological changes were examined after (1) hematoxylin and eosin (H&E) and (2) Periodic Acid Schiff (PAS) staining. Slides were examined by a blinded pathologist. Representative pictures are shown, magnification 10X (H&E) and 20X (PAS), respectively. Results represent at least two independent experiments and are expressed as the mean ± SEM (*n* = 10–24 per group). (\*) $P<0.05$, (\*\*) $P<0.01$, (\*\*\*) $P<0.001$, and (\*\*\*\*) $P<0.0001$ vs. control; (+) $P<0.05$, (++) $P<0.01$, and (+++) $P<0.001$ vs. TNBS group.

inflammation, edema, hyperplasia, and goblet cell depletion were significantly less severe in contrast to animals of the TNBS group. Additionally, the histological improvement was accompanied by a significant reduction of infiltration of inflammatory cells. MPO activity was reduced by over 75% (Fig 2F, $P<0.05$).

The PAS staining technique was employed to confirm the beneficial effect of sucrose esters in goblet cell recovery and their mucin production. In the TNBS group, the proportion of goblet cells containing neutral mucins (PAS-positive cells) was lower than in control animals ($P<0.0001$). We observed that PAS-positive cells were larger, nearly absent at the surface epithelium, and strongly reduced in areas where crypt abnormalities were more severe. In contrast, rats treated with the sucrose esters mixture from *P. peruviana* (5 and 10 mg/Kg) showed a remarkable increase of PAS-positive cells by over 2-fold ($P<0.05$ vs. TNBS group) (Fig 2G). The increased number of PAS-positive cells suggests that Peruvioses A and B mixture promotes tissue repair mechanisms.

## 3.2 Peruvioses A and B mixture showed a safe toxicological profile in rats treated with TNBS

A follow-up of the animals treated with sucrose esters from *P. peruviana* calyces, in both preventive and therapeutic experiments, allowed us to evaluate the safety of tested compounds. Within ~1–2 min after administration of the mixture of Peruviose A and B (20 mg/Kg/day; ip), animals assumed a recumbent posture and exhibited piloerection and occasional abdominal writhes, that peaked 2–5 minutes after injection and declined abruptly after that. Besides these adverse effects of pain/distress, no mortality or signs of toxicity were observed when compounds were administered at this dose for five days. On the other hand, this reaction to test compounds was not observed with lower doses of sucrose esters (10 and 5 mg/Kg/day; ip) when administered consecutively either for 5 or 15 days.

During necropsy, a macroscopic examination did not show detectable changes in the morphology of the liver and kidneys due to the administration of Peruvioses A and B mixture (20, 10, or 5 mg/Kg) in both experiments. In addition, the histopathology evaluation did not reveal significant changes in the tissue architecture of these organs in comparison to the control group (see S1 Fig).

## 3.3 Peruviose A and B reduce inflammatory gene and protein expression induced by TNBS instillation

To characterize the colonic inflammatory status, we examined the expression of iNOS, COX-2, cytokines (IL-1β, IL-6, IL-10, IL-17, and TNF-α) as well as proteins involved in mucus/epithelium integrity (MUC-2, MUC-3, and TFF3).

In the preventive study, TNBS administration promoted a significant increase of iNOS, COX-2, IL-1β, IL-6, and IL-10 expression ($P<0.05$ vs. control group), as well as a major reduction in MUC-2 ($P<0.05$ vs. control group). The administration of sucrose esters mixture from *P. peruviana* inhibited dose-dependently the expression of iNOS and COX-2 ($P<0.05$) while failing to modulate changes in the rest of the evaluated genes (Fig 3A). On the other hand, ELISA analysis confirmed that TNBS boosted the production of IL-1β, IL-6, IL-10, and TNF-α and reduced IL-4 levels (Fig 3B, $P<0.001$ vs. control group). Additionally, it demonstrated that animals treated with sucrose esters from *P. peruviana* produced a significant reduction of TNF-α and IL-10 (Fig 3B; $P<0.05$ vs. TNBS group), whereas the overproduction of other cytokines was not modified.

As can be seen in Fig 4A and 4B, the colonic inflammation induced by TNBS in the therapeutic setup was characterized by a prominent up-regulation of mRNA expression of iNOS, COX-2, IL-1β, IL-6, IL-10, and IL-17A ($P<0.05$ vs. control group). As expected, treatment with Peruvioses A and B mixture (5 and 10 mg/Kg/day, ip) strongly inhibited the expression of iNOS and COX-2 compared to TNBS rats ($P<0.05$). When the production of NO in colon tissue homogenates was evaluated (Fig 5B), we found that the treatment with sucrose esters

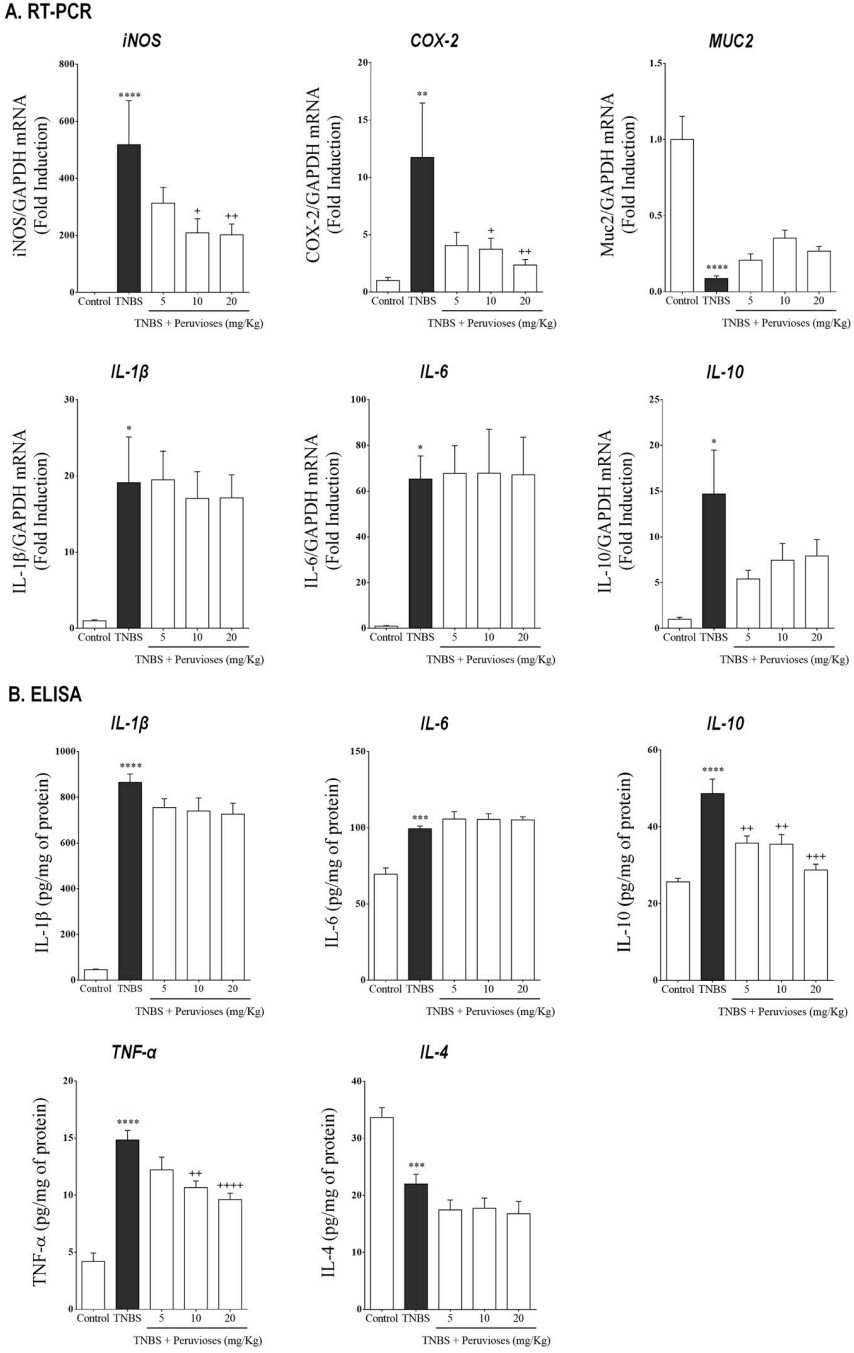

**Fig 3. Peruvioses A and B pre-treatment reduce the expression of pro-inflammatory enzymes (iNOS, COX-2) and cytokines (TNF-α, IL-10) in acute TNBS-colitis.** Wistar rats were treated with the Peruvioses mixture (5, 10, and 20 mg/Kg/day, ip) or vehicle for 2 days. Afterward, colitis was induced, as described in Materials and Methods. **(A)** iNOS, COX-2, MUC-2, IL-1β, IL-6, and IL-10 mRNA expression was quantified by RT-PCR. GAPDH was used as a housekeeping gene for normalization. **(B)** IL-1β, IL-6, IL-10, TNF-α, and IL-4 protein levels were measured by ELISA. Results represent at least two independent experiments and are expressed as the mean ± SEM ($n$ = 9–15 per group). (*) $P < 0.05$, (**) $P < 0.01$, (***) $P < 0.001$, and (****) $P < 0.0001$ vs. control; (+) $P < 0.05$, (++) $P < 0.01$, (+++) $P < 0.001$, and (++++) $P < 0.0001$ vs. TNBS group.

resulted in a marked decrease of the levels of this mediator, in comparison with the TNBS group (60% inhibition, $P<0.001$). Moreover, Western blot assays confirmed the reduction of iNOS expression at the protein level (Fig 5C) in rats treated with TNBS and the mixture of Peruvioses. On the other hand, as can be seen in the Fig 4A and 4B, the transcriptional levels of all the evaluated cytokines (IL-1β, IL-6, IL-10, and IL-17) were also significantly down-regulated in sucrose esters-treated animals ($P<0.05$ vs. TNBS group). Furthermore, ELISA analysis (Fig 5A) confirmed that these compounds were able to reduce the production of TNF-α, IL-1β, and IL-10 at the protein level ($P<0.05$ vs. TNBS group), whereas the levels of IL-6, IFN-γ, and IL-4 were not modified. Accordingly, the RT-PCR analysis (Fig 4D) showed a significant reduction in the expression of the nuclear transcription factor kappa B (NF-κB) in TNBS-treated rats with the mixture of Peruvioses ($P<0.05$ vs. TNBS group). This was verified at the protein level by western blot (Fig 5C).

TNBS prompted an important reduction in MUC-2 expression compared to control animals ($P<0.05$). Treatment with sucrose esters (5 and 10 mg/Kg) increased the expression of MUC-2 back to the levels of control animals (Fig 4C). Since the building block of intestinal mucus is MUC-2, a gel-forming mucin secreted by goblet cells, and taking into account our histological findings, we hypothesized that sucrose esters from *P. peruviana* might be stimulating the recovery of the colonic epithelium, thus enhancing goblet cells production of mucins. To test this, we further evaluated the expression of genes involved in mucus integrity. Peruvioses A and B demonstrated a beneficial impact in restoring the expression of TFF-3 ($P<0.05$) (Fig 4C). In the case of MUC-3, a tendency for higher expression in sucrose esters-treated animals was also observed.

## 3.4 Cytokine production is modulated by Peruvioses A and B in macrophages activated with LPS

In IBD patients, the proportion of lamina propria macrophages is increased, as well as the TLRs and NF-κB p65 expression, which is accompanied by increased production of IL-1β, IL-6, and TNF-α [32–35]. To elucidate the potential beneficial effect of sucrose esters from *P. peruviana* on macrophages, we investigated the response of RAW 264.7 cells to lipopolysaccharide (LPS) activation in the presence or absence of tested compounds (15–1 μg/mL). The mixture of Peruviose A and B significantly inhibited the LPS-induced production of NO, PGE2, IL-6, TNF-α, and MCP-1, in a concentration-dependent manner ($P<0.05$) without affecting cell viability as demonstrated by MTT assay (Fig 6). Surprisingly, IL-1β production was not modified by tested compounds (data not shown).

To verify that the bioactivity of sucrose esters was not related to an antioxidant effect, we evaluated the scavenging activity of compounds on ·NO, DPPH·, and ABTS$^{+•}$ radicals in a non-cellular system (see Supporting Information-Materials and Methods). Sucrose esters from *P. peruviana* did not exert an important scavenging effect, showing inhibition lower than 3% at 100 μg/mL (data not shown).

## 4. Discussion

The sucrose esters isolated from *P. peruviana* calyces have been recognized as promising anti-inflammatory agents [15]. Here, we show evidence of the beneficial effect of these sucrose esters in the TNBS-induced colitis model. Overall, our results suggest that ameliorating the altered immune response that characterizes the colonic inflammatory process induced by TNBS is involved in the favorable recovery of animals treated with these compounds.

The aerial surface of many genera of the Solanaceae family, such as *Datura*, *Lycopersicon*, *Nicotiana*, *Petunia*, *Physalis*, and *Solanum*, are covered by glandular trichomes, whose unique

## A. Pro-inflammatory Enzymes

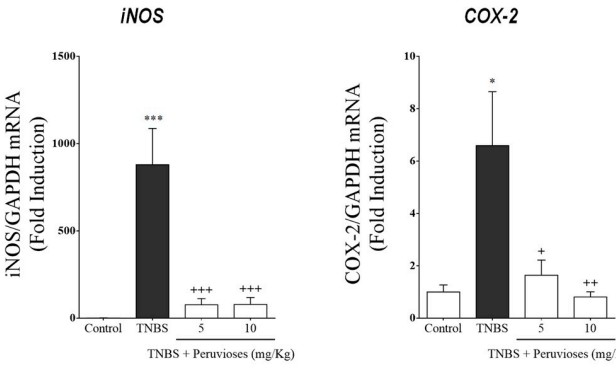

## B. Cytokines

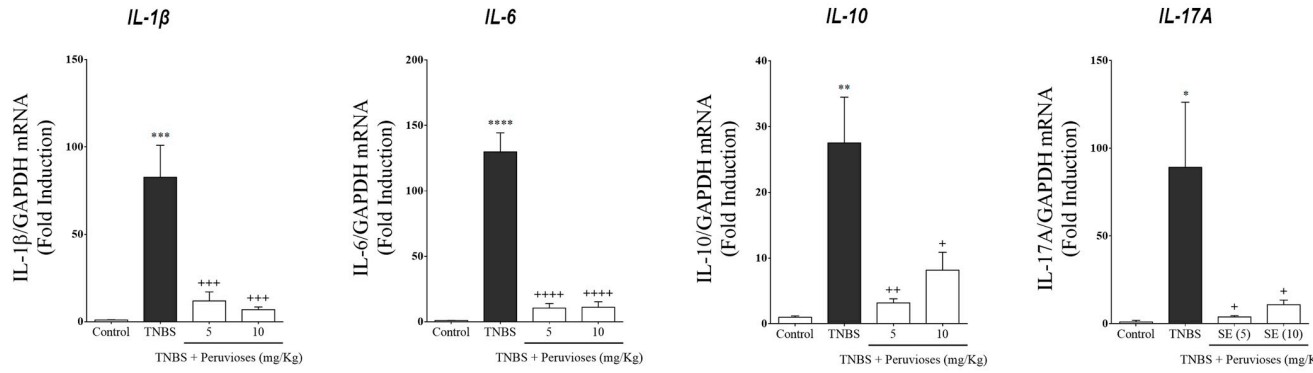

## C. Mucus Integrity

## D. Transcription Factor

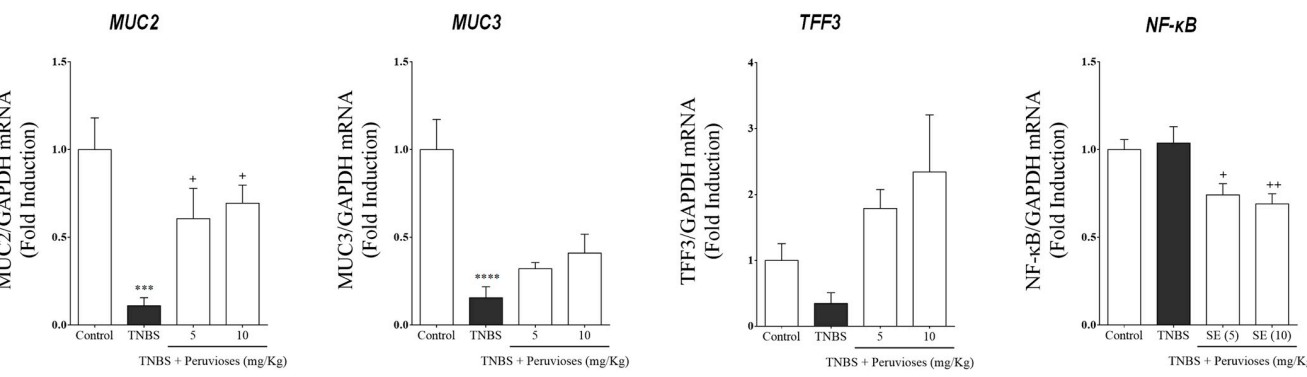

**Fig 4. Treatment with Peruvioses A and B modulates the expression of pro-inflammatory enzymes, cytokines, and NF-κB expression and restores markers of mucus integrity hampered in rats with established TNBS-induced colitis.** Wistar rats instilled with TNBS were treated with the Peruvioses mixture (5 and 10 mg/Kg/day, ip) or vehicle for two weeks, as described in Materials and Methods. **(A)** Pro-inflammatory enzymes (iNOS, COX-2), **(B)** Cytokines (IL-1β, IL-6, IL-10, and IL-17A), **(C)** Mucus integrity markers (MUC-2, MUC-3, and TFF-3), **(D)** and the transcription factor NF-κB mRNA expression was quantified by RT-PCR. GAPDH was used as a housekeeping gene for normalization. Results represent at least two independent experiments and are expressed as the mean ± SEM ($n$ = 7–10 per group). (*) $P<0.05$, (**) $P<0.01$, (***) $P<0.001$, and (****) $P<0.0001$ vs. control; (+) $P<0.05$, (++) $P<0.01$, (+++) $P<0.001$, and (++++) $P<0.0001$ vs. TNBS group.

and versatile secretory metabolism contributes to the wide variety of secondary metabolites of these plants [36, 37]. Glucose and sucrose esters have been characterized as the main compounds on the glandular trichome or its sticky exudate, which is believed to protect plants and fruits from insects and infections [38]. Previous studies have demonstrated that the *Physalis*

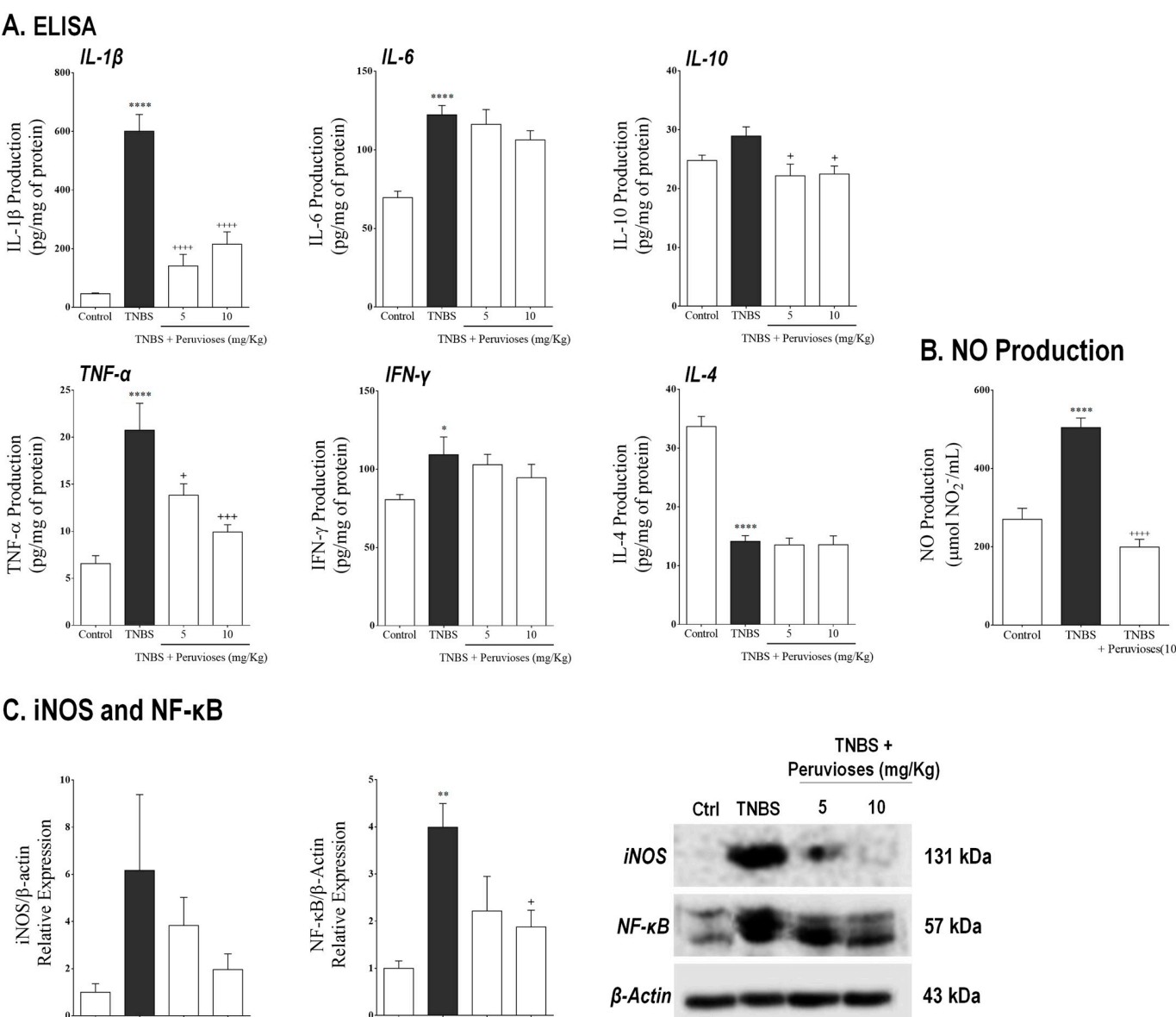

**Fig 5. Peruvioses A and B reduce NO and cytokines production probably by inhibition of NF-κB pathway.** Wistar rats instilled with TNBS were treated with the Peruvioses mixture (5 and 10 mg/Kg/day, ip) or vehicle for two weeks, as described in Materials and Methods. **(A)** Cytokines (IL-1β, IL-6, IL-10, TNF-α, IFN-γ, and IL-4) levels were measured by ELISA. **(B)** NO production was quantified employing the Griess reaction. **(C)** Expression of iNOS (cytoplasmic) and NF-κB (nuclear) were evaluated by immunoblotting. Results represent at least two independent experiments and are expressed as the mean ± SEM ($n$ = 7–10 per group). (\*) $P < 0.05$, (\*\*) $P < 0.01$, and (\*\*\*\*) $P < 0.0001$ vs. control; (+) $P < 0.05$, (+++) $P < 0.001$, and (++++) $P < 0.0001$ vs. TNBS group.

genus is an abundant source of sucrose esters [39, 40]. So far, twenty sucrose esters have been isolated from *P. viscosa*, *P. nicandroides* var attenuata, *P. sordida*, *P. solanaceus*, *P. peruviana*, *P. neomexicana*, and *P. philadelphica* [15, 41–46]. Among the pharmacological activities reported for these secondary carbohydrates, their anti-inflammatory effect appears to be the most important [40]. Particularly, we have focused on studying *P. peruviana* calyces, where sucrose esters are the most important metabolites.

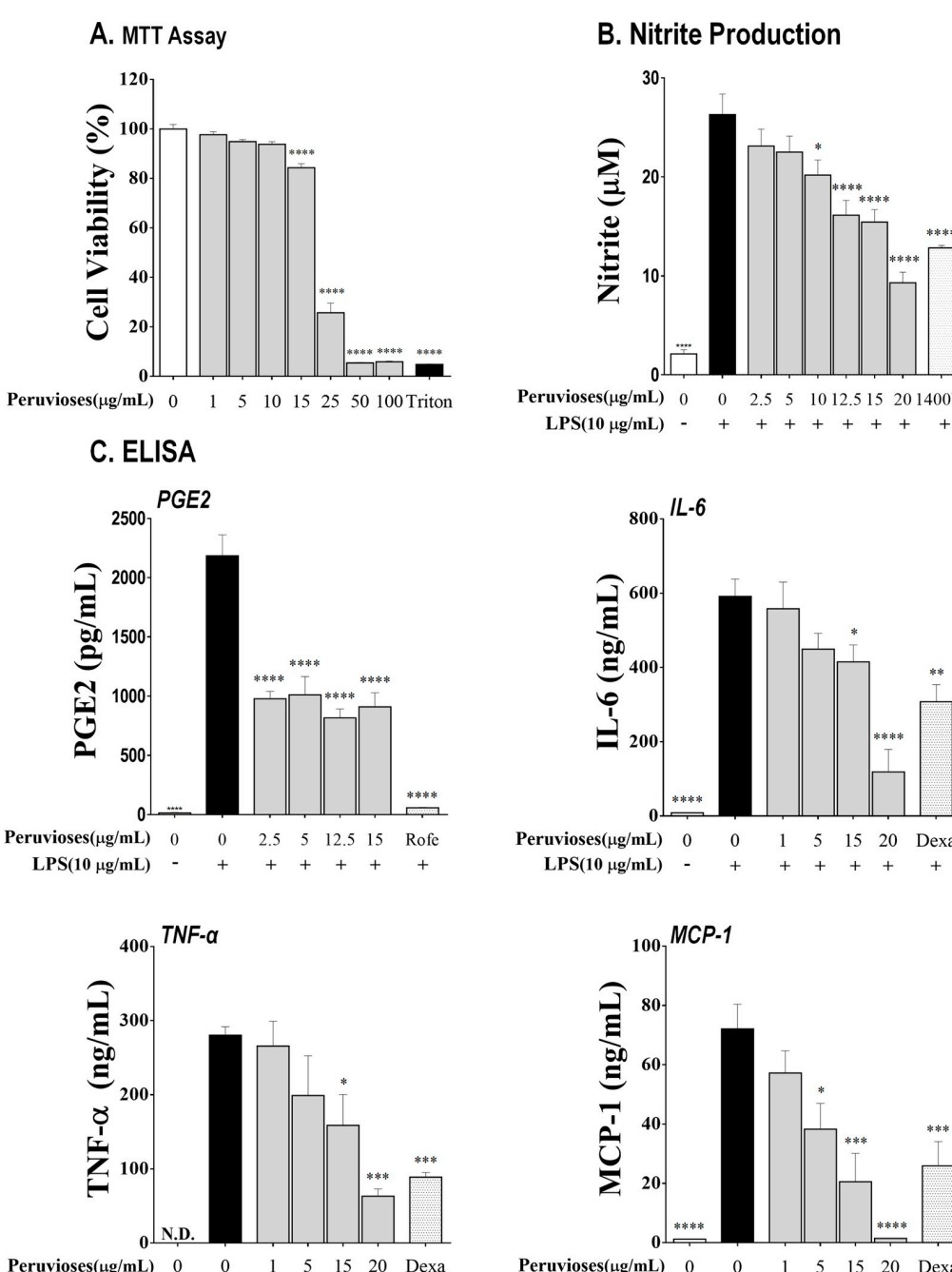

**Fig 6. Peruvioses A and B inhibited pro-inflammatory mediators production by LPS-stimulated RAW 264.7 cells.**
Macrophages were cultured with the Peruvioses mixture (0–100 μg/mL) for 30 minutes and then stimulated with LPS
(10 μg/mL) for 24 hours. **(A)** Cell viability was measured with the MTT assay, **(B)** Nitrite production was measured by
the Griess assay, and **(C)** PGE2, IL-6, TNF-α, and MCP-1 production were measured by ELISA. Results represent at
least three independent experiments and are expressed as the mean ± SEM (*) $P<0.05$, (**) $P<0.01$, (***) $P<0.001$, and
(****) $P<0.0001$ vs. control.

In this study, the Peruvioses A and B mixture was administered by intraperitoneal route.
The noxious response (writhing and piloerection) that occurred after the injection of test com-
pounds (20 mg/Kg/day) was not unexpected since we previously found this effect when

treating mice at higher doses [15]. However, the latency and magnitude of discomfort after administration with Peruvioses A and B at 20 mg/Kg was significantly less severe than the previously studied dosages (>100 mg/Kg), where acute toxicological effects of sucrose esters resulted in pronounced pain, piloerection, and hypoactivity [15]. This evidence indicates that sucrose esters produce local nociceptive effects in rodents.

In contrast, the 15-day administration of sucrose ester (5 and 10 mg/Kg) to Wistar rats showed no evidence of nociception/toxicity attributable to test compounds. Indeed, the histological examination revealed no changes in any of the animals treated with the Peruvioses when compared to the control group. Nevertheless, pain-related behavioral responses to intraperitoneal administration of sucrose esters remain surprising since carbohydrate polyesters, especially esters of sucrose and glucose, are commonly employed surfactants that have been approved as food, cosmetics, and pharmaceutical additives based on safety studies on animals and humans [47, 48]. In fact, sucrose esters have been proposed as emulsifiers, stabilizers, and absorption/penetration enhancers in drug delivery systems, especially for parenteral [49], nasal [50, 51], or topical administration [52]. Therefore, our data encourage further studies on their toxicological effect when administration routes differ from oral gavage. In the case of the Peruvioses mixture, additional research, including more extended administration periods, the inclusion of both male and female animals, and an assessment of its mutagenic and carcinogenic potential, should be performed to support their safety.

To evaluate the effect of the mixture of Peruviose A and B on intestinal inflammation, we employed the TNBS-induced colitis model. Among chemically induced models, TNBS remains as one of the most commonly used since it is considered a valuable tool to provide proof of concept for therapeutic interventions in a simple, time-saving, and relatively inexpensive setting [53–55]. Interestingly, many studies regarding complementary and alternative medicine on IBD have been done using this model [54, 56].

Although first developed for use in rats [25], this model has become widely used for inducing colitis in zebrafish [57–59], mice [60, 61], rabbits [62, 63], pigs [64–66], and dogs [67–69]. TNBS is a hapten administered as an enema in combination with ethanol (30–50%) to break the mucosal barrier and allow TNBS penetration into the bowel wall [55]. While some of the colonic damage is attributable to the caustic properties of the TNBS/ethanol solution [70], the inflammation is mainly induced by the haptenization of host proteins, which provokes severe ulcerations of the mucosal characterized by transmural infiltration of mononuclear cells [71, 72]. Results of this study demonstrated that Peruvioses A and B from *P. peruviana* were useful to ameliorate the damage induced in the colon of rats treated with TNBS either acutely (preventive set-up) or in an established disease (therapeutic set-up). In both settings, sucrose esters were able to improve the mucosal damage at macroscopic and histologic levels. However, the greatest beneficial effect of Peruvioses was noticed in the therapeutic set-up (two weeks of treatment), which indicates that a sufficient time of treatment is needed for sucrose esters to exert their effects fully.

Based on the time course of macroscopic, histologic, and immunological changes, inflammation by TNBS could be divided into several phases. The acute application of the TNBS-colitis model, where damage is assessed 3 days after TNBS instillation, is characterized by intense tissue injury and a non-specific inflammatory response [31]. In comparison, one week after TNBS administration, an established/chronic phase of inflammation sets in. In this phase, inflammation is characterized by a wide distribution of ulcers, cellular infiltration leading to granuloma formation, and a cytokine profile similar to Crohn's disease that peaks at two weeks [73]. In this sense, our data suggest the potential of sucrose esters from *P. peruviana* calyces to treat the chronic and persistent inflammation that characterizes IBD. In fact, in therapeutic set-up experiments, we observed that test compounds significantly accelerated the

recovery of TNBS-treated rats, reducing the mortality rate and weight loss, which was accompanied by reduced colon weight/length ratio, ulcer index, and damaged area extension. Regarding the histological analysis, our findings agreed with the previously reported changes induced by TNBS administration [25]. Furthermore, sucrose esters demonstrated a remarkable reduction of histological damage, consistent with the macroscopic observations.

Previous studies have established that the TNBS model shares many molecular features with human IBD, such as the overproduction of IFN-γ mediated by T helper 1 (Th1) T cells [74]. Although TNBS-induced cytokine secretion patterns are far more studied in mice, some studies have also verified a time-dependent changing pattern of Th1 cytokines in the rat model of TNBS, as well as important concordances with IBD transcriptomes [73, 75]. Consistently, we report that TNBS-treated rats exhibited a remarkable fold increase in iNOS, COX-2, IL-1β, IL-6, IL-10, and IL-17A expression, two weeks after TNBS instillation. ELISA analysis also demonstrated increased IFN-γ and TNF-α production, whereas IL-4 was diminished. Alternatively, the expression of markers of mucus/epithelial integrity (MUC-2, MUC-3, and TFF-3) were significantly impaired. Peruvioses A and B from *P. peruviana* were able to partially restore the inflammatory mediators and cytokine imbalance, with marked diminution of IL-1β, TNF-α, and iNOS at mRNA or protein levels. Moreover, sucrose esters normalized the expression of MUC-2 and TFF-3, which are core components of the mucus secreted by goblet cells [76]. Accordingly, we observed a higher number of goblet cells as evidenced by PAS-staining, which suggests that treatment with Peruvioses promoted epithelial regeneration in TNBS-treated rats.

Neutrophils recruitment to the intestine is a critical component of the inflammatory response. This process relies on selectins, integrins, and adhesion molecules for transendothelial and epithelial transmigration [77]. The association between the elevated expression of these molecules and intestinal inflammation has been well-documented in human and animal models of IBD [77]. For instance, the up-regulation of E-selectin (endothelial-selectin; CD62E); P-selectin (Platelet-selectin; CD62P); intercellular adhesion molecule 1 (ICAM-1); and vascular cell adhesion protein 1 (VCAM-1)] has been reported during TNBS-induced colitis in rats, peaking two weeks after induction of disease [73, 78, 79]. Indeed, it is known that their expression is regulated at the transcriptional level by TNF-α and IL-1β [78, 80, 81]. In agreement, our data showed that TNBS provoked abnormal recruitment of inflammatory cells to the colon tissue. Indeed, it is well established that this inflammatory infiltrate, mainly composed of neutrophils, is important for the development of colon edema [82]. From our point of view, the substantial reduction of TNF-α and IL-1β, both at the transcriptional and protein levels, caused by the sucrose esters from *P. peruviana* might be correlated to the significant decrease in MPO activity, which has been shown to be proportional to neutrophils infiltration in the colonic tissue [83].

Infiltrating macrophages play a crucial role in the pathogenesis of TNBS-induced colitis [84]. To clarify whether sucrose esters from *P. peruviana* could attenuate cytokine imbalance in macrophages, we employed LPS-stimulated RAW 264.7 cells. In the presence of the Peruvioses mixture, LPS-stimulated macrophages produced less NO, PGE2, IL-6, TNF-α, and MCP-1, at non-toxic concentrations. Alternatively, sucrose esters effects were unrelated to antioxidant properties, as demonstrated using DPPH and ABTS assays. Although the anti-inflammatory effect of plant-derived sucrose esters has been previously described, research has focused on COX-2 (PGE2) and iNOS (NO) inhibition [15, 45, 85], our results demonstrated that sucrose esters are able to modulate inflammation with a more complex mechanism of action. Nevertheless, since all the genes modulated by the Peruvioses A and B mixture, either *in vivo* or *in vitro*, are gene targets of the transcription factor NF-κB, we propose that the immunomodulatory effects of sucrose esters might result from the inactivation of NF-κB

signal pathway. Indeed, this study demonstrated that Peruvioses inhibited the mRNA expression and the nuclear translocation of NF-κB in rats treated with TNBS.

Although the experimental results from the TNBS-colitis model demonstrated the beneficial effect of sucrose esters from *P. peruviana*, several caveats are worth discussing. First, we only demonstrated the efficacy of Peruvioses A and B to treat colonic inflammation in one animal model, which does not entirely resemble IBD pathology in humans. Thus, the bioactivity of test compounds should be determined in at least another model. Additionally, TNBS-induced colitis, as applied in this work, is more representative of an acute IBD flare-up than a chronic condition; therefore, future studies should include an experimental setting of chronic inflammation to determine whether sucrose esters are helpful as long-term therapy for IBD [86]. On the other hand, given the chronic nature of IBD, the administration of test compound per oral should be preferred [71]; however, we employed intraperitoneal administration of Peruvioses. Although debatable, this administration route was chosen given that sucrose esters are metabolized in the gastrointestinal tract and are not absorbed [87]; thus, their bioavailability could be limited.

In summary, our results demonstrate that sucrose esters of *P. peruviana* calyces ameliorate the symptoms and progression of colitis in the TNBS-induced model by improving the epithelial recovery and modulating the cytokine unbalance associated with colitis, probably through suppression of NF-κB activation. Thus, sucrose esters from *P. peruviana* might be an attractive new complementary herbal alternative to treat IBD.

## Supporting information

**S1 Fig.** Treatment with Peruvioses A and B mixture (0–20 mg/Kg/day, ip) isolated from Physalis peruviana calyces did not produce an effect on the histological structure of liver and kidneys of TNBS-treated rats in the preventive (A) or therapy (B) set-up. Micrographs are representative of the histological section of organs stained with hematoxylin and eosin from at least six different animals. Magnifications 10X. Scores were assigned by a blinded pathologist according to the parameters established in S1Table, Supplementary Information. Each value represents the mean ± SEM.
(TIF)

**S2 Fig. [1]H-NMR spectra of Peruviose A and B from *P. peruviana*.**
(TIF)

**S3 Fig. FTIR spectra of Peruviose A and B mix from *P. peruviana*.**
(TIF)

**S1 Table. Scoring criteria of liver and kidney sections[a].**
(TIF)

**S2 Table. Sequences of primers used for Real-time PCR analysis.**
(TIF)

## Acknowledgments

The authors wish to thank Angelica Guerrero for her guidance in RT-PCR experiments and Lia Barrios for her collaboration in histopathology assays. We also acknowledge William Padilla, Marlon Quintana, Luis Barrios, Julio Acuña, and David Rivera for their collaboration during experiments.

## Author Contributions

**Conceptualization:** Luis A. Franco.

**Formal analysis:** Yanet C. Ocampo, Jenny P. Castro, Daneiva Caro, Elena Talero, Virginia Motilva, Luis A. Franco.

**Funding acquisition:** Luis A. Franco.

**Investigation:** Yanet C. Ocampo, Jenny P. Castro, Daneiva Caro.

**Methodology:** Yanet C. Ocampo, Jenny P. Castro, Elena Talero, Luis A. Franco.

**Project administration:** Luis A. Franco.

**Supervision:** Virginia Motilva, Luis A. Franco.

**Visualization:** Indira B. Pájaro.

**Writing – original draft:** Yanet C. Ocampo.

**Writing – review & editing:** Jenny P. Castro, Indira B. Pájaro, Daneiva Caro, Elena Talero, Virginia Motilva, Luis A. Franco.

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
