## [Decision Letter · Decision Letter 0]

19 Dec 2023

PONE-D-23-23074Protective effect of sucrose esters from Cape gooseberry (Physalis peruviana L.) in TNBS-induced colitisPLOS ONE

Dear Dr. Franco,

Thank you for submitting your manuscript to PLOS ONE. After careful consideration, we feel that it has merit but does not fully meet PLOS ONE’s publication criteria as it currently stands. Therefore, we invite you to submit a revised version of the manuscript that addresses the points raised during the review process.

We look forward to receiving your revised manuscript.

Kind regards,

Mathilde Body-Malapel

Academic Editor

PLOS ONE

Journal Requirements:

2. We note that this submission includes NMR spectroscopy data. We would recommend that you include the following information in your methods section or as Supporting Information files:

          1) The make/source of the NMR instrument used in your study, as well as the magnetic field strength. For each individual experiment, please also list: the nucleus being measured; the sample concentration; the solvent in which the sample is dissolved and if solvent signal suppression was used; the reference standard and the temperature.

          2) A list of the chemical shifts for all compounds characterised by NMR spectroscopy, specifying, where relevant: the chemical shift (δ), the multiplicity and the coupling constants (in Hz), for the appropriate nuclei used for assignment.

           3)The full integrated NMR spectrum, clearly labelled with the compound name and chemical structure.

We also strongly encourage authors to provide primary NMR data files, in particular for new compounds which have not been characterised in the existing literature. Authors should provide the acquisition data, FID files and processing parameters for each experiment, clearly labelled with the compound name and identifier, as well as a structure file for each provided dataset. See our list of recommended repositories here: " ext-link-type="uri" xlink:type="simple">https://journals.plos.org/plosone/s/recommended-repositories"

Reviewers' comments:

Reviewer's Responses to Questions

**Comments to the Author**

1. Is the manuscript technically sound, and do the data support the conclusions?

Reviewer #1: Yes

Reviewer #2: Yes

2. Has the statistical analysis been performed appropriately and rigorously? 

Reviewer #1: Yes

Reviewer #2: Yes

3. Have the authors made all data underlying the findings in their manuscript fully available?

Reviewer #1: Yes

Reviewer #2: Yes

4. Is the manuscript presented in an intelligible fashion and written in standard English?

Reviewer #1: Yes

Reviewer #2: Yes

5. Review Comments to the Author

Reviewer #1: Introduction: Line 87. Remoe the accent sign in the word Peru.

Methods: 2.1 Plant material. Add lattitude and longitude information. Add date collection.

The legends are not self-explaied. They should be more elaborated.

Reviewer #2: Dear Authors,

The study is very current. IBD is a big problem in human medicine. The disease affects people in North America and in Europe. Many researchers look at new drugs and/or substances to treat these diseases. They are looking for substances of a natural origin, from plants. Phytotherapy is an attractive strategy to treat IBD. In my opinion a perfect idea. Many scientific articles described plant substances as a new method for, among other, civilizations diseases. These included bowel disease -ulcerative colitis (IC) and Crohn`s disease (CD). This well-written manuscript is suitable for publication after minor revision, after the question answers.

In this study, the Authors describe sucrose esters - peruvioses A and B from Physalis peruviana in different doses as substances with beneficial effects for TNBS-induced colitis All described methods and results with figures, bar graphs, and microphotographs are legible and clear. In the manuscript is a lack of scale bar in microphotographs of intestine. In supplementary file in description or on S1 Fig 1 is a lack of magnification or scale bar.

The results and conclusions are very attractive but with some restrictions – the sucrose esters from Physalis peruviana as an alternative in IBD treatment. The intraperitoneal dosage must be determined in another model and/or in another study and animal species. Are the intraperitoneal injections safe for long-term treatment?

Best regards

6. PLOS authors have the option to publish the peer review history of their article (what does this mean?). If published, this will include your full peer review and any attached files.

Reviewer #1: **Yes: **Amílcar Sabino Damazo

Reviewer #2: No

---

## [Author Response · Author response to Decision Letter 0]

1 Feb 2024

Cartagena, January 29th 2024

Doctor

Emily Chenette

Editor-in-chief

Plos-One 

Dear Dra. Chenette, we are resubmitting our manuscript, “Protective effect of sucrose esters from Cape gooseberry (Physalis peruviana L.) in TNBS-induced colitis” by Yanet Ocampo et al. I inform you that we have accepted the changes suggested by the referees and the editorial office, and prepared a new version of our manuscript. Following, we detailed the responses to the reviewers' comments and questions:

Editor

Comment 1: Please ensure that your manuscript meets PLOS ONE's style requirements, including those for file naming.

We revised and corrected the manuscript as suggested by the editorial office according to Plos One style templates.

Comment 2: We note that this submission includes NMR spectroscopy data. We would recommend that you include the following information in your methods section or as Supporting Information files.

1) The make/source of the NMR instrument used in your study, as well as the magnetic field strength. For each individual experiment, please also list: the nucleus being measured; the sample concentration; the solvent in which the sample is dissolved and if solvent signal suppression was used; the reference standard and the temperature.

This information was included In Section 1.5 of the Supplementary Information.

2) A list of the chemical shifts for all compounds characterized by NMR spectroscopy, specifying, where relevant: the chemical shift (δ), the multiplicity and the coupling constants (in Hz), for the appropriate nuclei used for assignment.

3) The full integrated NMR spectrum, clearly labelled with the compound name and chemical structure.

Full integrated NMR spectrum was included in S2 Fig. Supplementary Information. On the other hand, about the suggestion to include the list of the chemical shifts for all compounds characterized by NMR spectroscopy, specifying, where relevant, the chemical shift (δ), the multiplicity, and the coupling constants (in Hz) for the appropriate nuclei used for assignment, we believe it is neither necessary nor proper to include this information since this data was published in Planta Med 2014; 80(17): 1605-1614 DOI: 10.1055/s-0034-1383192, including the compound's molecular structures.

Nevertheless, to provide clarity on the comparative structural analysis performed in this work, we include below a comparison of the two spectra, the one previously published and the one taken to ensure the identity of Peruviose A and B:

As can be observed, the spectra are practically identical; the only differences correspond to a residue of ethyl acetate, which was the elution solvent of the chromatographic column used for the separation and purification of Peruviose A and B. Additionally, the absence of a signal between 1.5 and 2.0 ppm corresponding to residual water of the deuterated solvent can be observed, probably because a deuterated solvent without humidity was used on this occasion.

We also strongly encourage authors to provide primary NMR data files, in particular for new compounds which have not been characterised in the existing literature. Authors should provide the acquisition data, FID files and processing parameters for each experiment, clearly labelled with the compound name and identifier, as well as a structure file for each provided dataset. See our list of recommended repositories here: https://journals.plos.org/plosone/s/recommended-repositories"

According to your suggestion, we include the FID file obtained in the repository of our data.

Comment 3: Note from Emily Chenette, Editor in Chief of PLOS ONE, and Iain Hrynaszkiewicz, Director of Open Research Solutions at PLOS: Did you know that depositing data in a repository is associated with up to a 25% citation advantage (https://doi.org/10.1371/journal.pone.0230416)? If you’ve not already done so, consider depositing your raw data in a repository to ensure your work is read, appreciated and cited by the largest possible audience. You’ll also earn an Accessible Data icon on your published paper if you deposit your data in any participating repository (https://plos.org/open-science/open-data/#accessible-data).

All data from the experiment, including photos, tables, spectra, and all files related to the research, are incorporated in the repository of the University of Cartagena and can be accessed through this link or QR code: https://repositorio.unicartagena.edu.co/handle/11227/5014.

We checked our reference list, ensuring that it is complete and correct. Additionally, it was adjusted to the format of the journal.

Reviewers' comments:

Reviewer 1 - Comment 1: Introduction: Line 87. Remove the accent sign in the word Peru.

The accent in the word Peru was removed. Line 49 in the Revised Manuscript with Track Changes.

Reviewer 1 - Comment 2: Methods: 2.1 Plant material. Add latitude and longitude information. Add date collection.

The information on latitude, longitude, and date collection was included. Line 88 in the Revised Manuscript with Track Changes.

Reviewer 1 - Comment 3: The legends are not self-explained. They should be more elaborated.

All legends were revised, and we consider they contain the necessary information to be self-explained. No modifications were made.

Reviewer 2 - Comment 1: The study is very current. IBD is a big problem in human medicine. The disease affects people in North America and in Europe. Many researchers look at new drugs and/or substances to treat these diseases. They are looking for substances of a natural origin, from plants. Phytotherapy is an attractive strategy to treat IBD. In my opinion a perfect idea. Many scientific articles described plant substances as a new method for, among other, civilizations diseases. These included bowel disease -ulcerative colitis (IC) and Crohn`s disease (CD). This well-written manuscript is suitable for publication after minor revision, after the question answers. In this study, the Authors describe sucrose esters - peruvioses A and B from Physalis peruviana in different doses as substances with beneficial effects for TNBS-induced colitis All described methods and results with figures, bar graphs, and microphotographs are legible and clear. In the manuscript is a lack of scale bar in microphotographs of intestine. In supplementary file in description or on S1 Fig 1 is a lack of magnification or scale bar.

Figures 1 and 2 were adjusted, including a ruler in centimeters in microphotographs of the intestine. 

In S1 Fig, the magnification used was 10X, and it is included in the corresponding legend. 

Reviewer 2 - Comment 2: The results and conclusions are very attractive but with some restrictions – the sucrose esters from Physalis peruviana as an alternative in IBD treatment. The intraperitoneal dosage must be determined in another model and/or in another study and animal species. Are the intraperitoneal injections safe for long-term treatment?

We agree with the reviewer's assessment, and therefore, the safety of the sucrose esters (Peruviose A and B) was tested at doses of 2.5, 5, and 10 mg/Kg/day for 28 consecutive days in mice of both sexes administered intraperitoneally following the Guidelines for the Testing of Chemicals from the Organization for Economic Cooperation and Development - OECD - Test 407. This study was published in 2017 and is available in Biomedicine Pharmacotherapy, 90, 850-862. https://doi.org/10.1016/j.biopha.2017.04.046

Regarding whether intraperitoneal injections are safe for long-term treatment?

Repeated intraperitoneal administration of substances may produce non-specific adverse effects, which may be due to the substance administered, the vehicle used, or the administration process, so this administration in humans should be avoided. In experimental animals and especially rodents, the intraperitoneal route is widely used as a route of drug administration due to its simplicity, rapid absorption, and high bioavailability. This route was chosen in our case since sucrose esters (Peruviose A and B) are degraded orally. Considering the limitations of this route of administration (IP), we are currently working on developing a pharmaceutical formulation that allows oral administration, protecting the Peruvioses from gastrointestinal conditions.

Sincerely,

Prof. LUIS FRANCO OSPINA, Ph.D.

Corresponding author

Biological Evaluation Group Director 

Universidad de Cartagena - Colombia.

---

## [Decision Letter · Decision Letter 1]

15 Feb 2024

Protective effect of sucrose esters from Cape gooseberry (Physalis peruviana L.) in TNBS-induced colitis

PONE-D-23-23074R1

Dear Dr. Franco,

We’re pleased to inform you that your manuscript has been judged scientifically suitable for publication and will be formally accepted for publication once it meets all outstanding technical requirements.

Kind regards,

Mathilde Body-Malapel

Academic Editor

PLOS ONE

Additional Editor Comments (optional):

Reviewers' comments:

Reviewer's Responses to Questions

**Comments to the Author**

1. If the authors have adequately addressed your comments raised in a previous round of review and you feel that this manuscript is now acceptable for publication, you may indicate that here to bypass the “Comments to the Author” section, enter your conflict of interest statement in the “Confidential to Editor” section, and submit your "Accept" recommendation.

Reviewer #1: All comments have been addressed

Reviewer #2: All comments have been addressed

2. Is the manuscript technically sound, and do the data support the conclusions?

Reviewer #1: Yes

Reviewer #2: Yes

3. Has the statistical analysis been performed appropriately and rigorously? 

Reviewer #1: Yes

Reviewer #2: Yes

4. Have the authors made all data underlying the findings in their manuscript fully available?

Reviewer #1: Yes

Reviewer #2: Yes

5. Is the manuscript presented in an intelligible fashion and written in standard English?

Reviewer #1: Yes

Reviewer #2: Yes

6. Review Comments to the Author

Reviewer #1: All suggestions have been adressed. No further questions are needed. I agree with the publication of the manuscript.

Reviewer #2: Dear Authors,

Thank You very much for all corrections, answers and scale bar in figures.

In my opinion, this manuscript well-written after the corrections is suitable for publication

Best regards

7. PLOS authors have the option to publish the peer review history of their article (what does this mean?). If published, this will include your full peer review and any attached files.

Reviewer #1: **Yes: **Amílcar Sabino Damazo

Reviewer #2: No

---

## [Editor Report · Acceptance letter]

13 Mar 2024

PONE-D-23-23074R1 

PLOS ONE

Dear Dr. Franco, 

I'm pleased to inform you that your manuscript has been deemed suitable for publication in PLOS ONE. Congratulations! Your manuscript is now being handed over to our production team.

Kind regards, 

on behalf of

Dr. Mathilde Body-Malapel 

Academic Editor

PLOS ONE